# Exposure Media and Nanoparticle Size Influence on the Fate, Bioaccumulation, and Toxicity of Silver Nanoparticles to Higher Plant *Salvinia minima*

**DOI:** 10.3390/molecules26082305

**Published:** 2021-04-16

**Authors:** Melusi Thwala, Stephen Klaine, Ndeke Musee

**Affiliations:** 1Water Centre, Council for Scientific and Industrial Research, Pretoria 0001, South Africa; mthwala@csir.co.za; 2Zoology Department, University of Johannesburg, Auckland Park 2006, South Africa; 3Department of Environmental Health, Nelson Mandela University, Port Elizabeth 6031, South Africa; 4Centre for Environmental Management, University of the Free State, Bloemfontein 9300, South Africa; 5Department of Biological Sciences, Clemson University, Clemson, SC 29634, USA; sklaine@clemson.edu; 6Emerging Contaminants Ecological and Risk Assessment (ECERA) Research Group, Department of Chemical Engineering, University of Pretoria, Pretoria 0001, South Africa

**Keywords:** silver nanoparticles, bio-interaction, nanoecotoxicity, aquatic plant, bioaccumulation

## Abstract

Silver nanoparticles (AgNPs) are favoured antibacterial agents in nano-enabled products and can be released into water resources where they potentially elicit adverse effects. Herein, interactions of 10 and 40 nm AgNPs (10-AgNPs and 40-AgNPs) with aquatic higher plant *Salvinia minima* at 600 µg/L in moderately hard water (MHW), MHW of raised calcium (Ca^2+^), and MHW containing natural organic matter (NOM) were examined. The exposure media variants altered the AgNPs’ surface properties, causing size-dependent agglomeration. The bio-accessibility in the ascending order was: NOM < MHW < Ca^2+^, was higher in plants exposed to 10-AgNPs, and across all exposures, accumulation was higher in roots compared to fronds. The AgNPs reduced plant growth and the production of chlorophyll pigments *a* and *b*; the toxic effects were influenced by exposure media chemistry, and the smaller 10-AgNPs were commonly the most toxic relative to 40-AgNPs. The toxicity pattern was linked to the averagely higher dissolution of 10-AgNPs compared to the larger counterparts. The scanning electron microscopy and X-ray fluorescence analytical techniques were found limited in examining the interaction of the plants with AgNPs at the *low* exposure concentration used in this study, thus challenging their applicability considering the even lower predicted environmental concentrations AgNPs.

## 1. Introduction

Silver engineered nanoparticles (AgNPs) are widely used in nano-enabled products (NEPs), including personal care products, fabrics, disinfectant sprays, vacuum cleaners and air conditioners [1,2,3], chiefly due to their antibacterial properties [4,5,6]. AgNPs are among low volume production engineered nanomaterials (ENMs) with global production of 135 to 420 tons by 2015 [7], are not the highest applied ENMs in NEPs [1,2,3]; however, they are expected to reach *ca* 800 tons by 2025 [8] owing to the high market value compared to other ENMs [3]. 

Thus, the rising use of AgNPs has led to their increasing emission into water resources, and at present ranked among the top five ENMs emitted from NEPs into the aquatic environments [3]. In turn, there is increasing evidence on the environmental occurrence of AgNPs with concentrations in surface water in the range 3^−5^–2.63 µg/L [9,10,11], and expected to increase as more NEPs are introduced into the market. Yet, concerns related to their potential deleterious effects and environmental implications remain poorly quantified, especially to aquatic higher plants [12]. To date, the toxicity of AgNPs to aquatic biota has been extensively studied—with the basic toxicity mechanism mostly ascribed to the release of Ag^+^ ions and well-documented for microorganisms and daphnids [13,14,15] but to a lesser extent to higher plants [12,16,17]. Several studies have investigated the interaction of metal-based ENMs with aquatic higher plants [12,17,18,19,20,21,22], but gaps still exist with reference to the interactive influence of the ENMs’ and environmental physio-chemical factors on bio-interaction and toxicity to higher plants. 

Few studies reporting on the toxicity of ENMs have identified ENMs’ size [23,24,25,26], surface properties [27,28,29], dissolution [17,27,28], or exposure period [17,20] as influencing factors on their bioavailability, uptake, and toxicity effects to higher plants. For example, the size of AgNPs significantly influences their toxicity to aquatic higher plants, with smaller forms being more toxic than the larger ones irrespective of dissolution rates [17,24,26]. Further, the accumulation of either dissolved or nano-particulates forms by higher plants has been ascribed to adsorption and absorption processes predominantly via the roots where the former are easily internalised compared to the latter [29,30,31], although a reverse trend on internalisation is also possible [32]. Comparative studies on aquatic higher plants’ accumulation of Ag from Ag salts and AgNPs exposures have demonstrated higher accumulation from the former relative to the latter due to high dissolution of the salts [12,17,33]. However, remarkably in certain instances, the effect of dissolution can be insignificant [29].

Therefore, Ag salts tend to be more toxic to aquatic higher plants compared to AgNPs mainly due to enhanced uptake of dissolved Ag [29,33], although there are arguments that at similar uptake rates, AgNPs could be more toxic [33]. However, the latter hypothesis remains untested and highlights the need for detailed interaction assessments between aquatic higher plants and AgNPs. Whereas dissolution plays a significant role in the behaviour and effects of metal-based ENMs in aquatic environments, bio-mediated physico-chemical changes may also influence their uptake. For example, Stegemeier et al. [29] observed that AgNPs in contact with duckweed *Landoltia punctuta* were bio-transformed into AgS and Ag-thiol within tissues. 

Thus, although dissolution driven toxicity of AgNPs cannot be dismissed, it can also occur even under a negligible dissolution state [22]. Broadly, it is evident that the physico-chemical transformations exhibited by AgNPs determine their uptake and toxicity dynamics; hence, raising the need for robust data to demonstrate how such drivers are linked. At present, water chemistry is established as a key determinant of ENMs behaviour and toxicity in aquatic environments [33,34,35], but it remains unclear how it influences the interaction and bioavailability dynamics of ENMs to aquatic higher plants [12].

The aim of the present study was to investigate the interaction, uptake, accumulation, and toxic effects of citrate coated 10 nm (10-AgNPs) and 40 nm AgNPs (40-AgNPs) to *Salvinia minima* under different exposure water chemistry conditions. Although previously similar studies have been reported for daphnids [33,34,35]; to date, to the best of our knowledge, only one study has been reported for the aquatic higher plants [17,36]. In this work, *Salvinia minima* was chosen as a test model for aquatic higher plants because it can be cultivated under laboratory conditions with relative ease, exhibits rapid growth and metal bioaccumulation rates, and provides sufficient plant biomass necessary for ecotoxicological assessments relative to smaller counterparts such as *Lemna* and *Spirodela* sp.

## 2. Methods

### 2.1. Engineered Nanoparticles and Chemical Consumables

Stocks of two citrate stabilised AgNPs suspensions (0.02 mg/mL) were purchased from Sigma-Aldrich, St. Louis, MO, USA, with average sizes of 10 nm (10-AgNPs, ID catalogue 730785-25 mL) and 40 nm AgNPs (40-AgNPs, ID catalogue 730807-25 mL) according to the supplier’s specifications. In the laboratory, the batches of ENMs were stored at 4 °C in darkness prior to use. Hoagland’s No. 2 Basal salt mixture (Appendix A) was obtained from Sigma Aldrich, USA (Catalog No.: H2395) and used to prepare Hoagland’s medium (HM) in deionised water (DIW) (15 mΩ). The Suwannee River natural organic matter standard (reference 1R101N) was used. Moderately hard water (MHW) was used as the exposure media and prepared following the US EPA standard formulation [37]: 96 mg/L NaHCO_3_, 60 mg/L CaSO_4_, 60 mg/L MgSO_4_, and 4 mg/L KCl in DIW.

### 2.2. Exposure Suspensions

The MHW was the exposure media, and consisted of three variants. The first variant was the standard MHW described in Section 2.1 herein referred to as MWH exposure. The second variant contained 5 mg/L Suwanee River natural organic matter (herein NOM exposure), and the third one contained additional calcium to 90 mg/L (herein Ca^2+^ exposure). Following 24 h stabilisation of each exposure media, a nominal exposure dose of 600 µg/L for each AgNPs size was prepared in the three exposure media variants. Multiple exposure concentrations were not examined because primarily the study intended to examine AgNPs media transformation, biota interaction and effects in *Spirodela*, the assessment of dose-response dynamics was not the objective of the study. All exposure suspensions were bath-sonicated (Fischer Scientific, FS30) for 30 min, followed by pH adjustment to 8.0 ± 0.2 before experimental initiation as initially, the various media exhibited pH values that were either closely below or above 8.

### 2.3. Characterisation of AgNPs

Before initiating the experiments, the size and morphology of AgNPs were determined using the Transmission electron microscopy (TEM; JEM 2100, JEOL, Tokyo, Japan), dynamic light scattering technique (DLS; Zetasizer Nano ZS, Malvern Instruments, Malvern, UK), and the nanoparticle tracking analysis (NTA; NS500, NanoSight, Malvern, UK). Additionally, DLS was used to determine the AgNPs’ zeta potential, NTA to determine the particles/mL concentration. For TEM analysis, 100× dilution of the stock suspensions with DIW was undertaken in 2 mL microtubes, vortexed for 1 min, bath-sonicated for 5 min, followed by dipping of the copper (Cu) TEM grid in the sample. The grids were covered and left to air dry before analysis. Herein the reported average size was obtained from a minimum of 10 counts per image out of five images per sample. For NTA and DLS analysis, 1000× dilutions of stocks were prepared in 15 mL microtubes, vortexed for 1 min, bath-sonicated for 5 min before the analysis was undertaken. All analysis was done in triplicates. Modal and average sizes are reported for NTA and Zetasizer, respectively, and results are listed in Table 1. 

### 2.4. Laboratory Maintenance of Plant Culture

A laboratory culture of *S. minima* was initiated at Clemson University Institute of Environmental Toxicology laboratories (CU-ENTOX) with individuals obtained from stocks in the Biological Sciences Department Green House, University of Clemson, USA. On arrival, the plants were rinsed with tap water to remove attached debris, followed by culture initiation in 10% HM. The plants were maintained in 10 L glass tanks in a REVCO incubation chamber at 24 °C 16 h light and 8 h dark photoperiod, and under white light for at least 7 d before they were exposed to AgNPs. 

### 2.5. Bioaccumulation, Uptake and Distribution Experiments

Healthy non-chlorotic plants with intact roots were harvested from the stock plant cultures and blot dried on a paper towel. All exposure experiments were conducted in triplicate, and each replicate contained 500 mg fresh weight biomass test population. The plants were exposed to 50 mL of 600 µg/L AgNPs suspension in 125 mL volumetric flasks for 48 h at 24 °C 16 h light and 8 h dark photoperiod, with 24 h time points. To reduce evaporation and also facilitate gaseous exchange, the test vessels were covered with pierced parafilm. After 48 h, the exposed plants were harvested and gently rinsed with DIW, dried on a paper towel, followed by weighing to determine the fresh weight. Next, the roots and fronds were separated, dried at 80 °C for 4 h in 20 mL crucibles, and the dry tissue weight was determined after cooling. The obtained dry biomass was heated at 530 °C for 14 h in a furnace (Fischer Scientific, Waltham, MA, USA), and the resultant ash prepared for total Ag analysis. 

### 2.6. Electron Microscopy and X-ray Fluorescence Spectroscopy

To visualise the plants-AgNPs’ interactions, the exposure protocol already described in Section 2.5 was extended to 72 h. After 72 h, the plants were rinsed with DIW, followed by separation of the roots from fronds. The roots were covered in foil, snap-frozen in liquid nitrogen, and stored at −80 °C prior to further processing. In preparation for analysis, samples were removed from the freezer and freeze-dried under vacuum (under conditions in Appendix A). After freeze-drying, sections of roots were fixed on stubs and viewed under the scanning electron microscope (SEM) (JOEL JSM 7500F, Tokyo, Japan) using the secondary electron (SE) detector at an acceleration voltage of 2 kV coupled with Energy dispersive X-ray (EDX) spectroscopy operated at 15 kV for elemental analysis, and analysis was triplicated. 

To examine the distribution of Ag in the roots following plants exposure to AgNPs, the energy-dispersive micro-X-ray fluorescence spectrometer (μ-EDXRF) was used. Samples were obtained at 24, 48, and 72 h, rinsed in DIW and mounted on 7 mm thick clear Perspex plastic, and air-dried for 48 h at 60 °C in an oven (ESCO, Isotherm^®^, Singapore). After sample drying, the top layer of the Perspex was removed, and elemental mapping was undertaken at 25 and 50 keV, 500 μA; the beam focused on <25 μm spot sizes with 51° incident beam and take-off angles. 

### 2.7. Total and Dissolved Ag Analysis

Inductively coupled plasma mass spectroscopy (ICP-MS; Agilent 7500 Series, Santa Clara, CA, USA) was used to measure dissolved Ag in the exposure media at 24 h intervals. The samples were prepared by ultracentrifugation with 3 kDa Amicon^®^ filter units, and 5% HNO_3_ acidification. Total Ag associated with the exposed plants was measured after 48 h. The obtained plant ash (Section 2.5) was digested using 250 µL HNO_3_ until fully dissolved, then DIW added to achieve a 5% acid concentration. The resultant aqueous suspension was centrifuged at 3880 rpm for 10 min before total Ag determination was done in triplicates. Overall, recoveries in the range of 78–113% were achieved (Appendix A).

### 2.8. Growth Assay 

Growth assays were initiated using 350 mg fresh plant biomass exposed to 600 µg/L AgNPs in MHW, NOM, and Ca^2+^ exposure media variants for 7 d under incubation parameters described in Section 2.5. All exposures were done in triplicate. On day 7 the plants were harvested, and fresh biomass measured after paper blot drying. Relative growth rate (RGR) was determined following expression [27]
(1)$$\mathrm{RGR}\;=\;\frac{lnW_{2}\;-\;lnW_{1}}{t}$$
where *W*_1_ and *W*_2,_ respectively, represent the initial and final fresh weight (mg) and *t* the incubation time (d).

### 2.9. Chlorophyll Pigments Assay

Photosynthetic effects were determined following the protocol described in Section 2.5. In brief, immediately after obtaining the whole plant biomass on day 7, the roots and fronds were separated, and thereafter the fronds’ fresh biomass was determined. Next, the chlorophyll pigments were extracted using 5 mL 96% ethanol after homogenisation with a pestle and mortar. The mixture was centrifuged at 3880 rpm for 10 min before the absorbance (*A*) of the supernatant was read at λs of 470, 649, and 664 nm using the ultraviolet-visible spectrophotometer (Shimadzu, UV-2501PC, Kyoto, Japan). The concentrations of the pigments were calculated following the protocol described by [38] using the expressions
(2)$${\mathrm{Chl}}_{a}\;=\;\frac{\left(13.36\cdot A_{664}\;-\;5.19\cdot A_{649}\right)\cdot 8.1}{FW}$$
(3)$${\mathrm{Chl}}_{b}\;=\;\frac{\left(27.43\cdot A_{649}\;-\;8.12\cdot A_{664}\right)\cdot 8.1}{FW}$$
where Chl*_a_* and Chl*_b_*, respectively, are chlorophylls *a* and *b*, $A_{\lambda }$ is the absorbance at a given *λ*, and *FW* is the fresh weight. 

### 2.10. Statistical Analysis

All tests were performed in triplicate, and the results herein are presented as means plus standard deviations. Differences between treatments (>2) were tested using JMP Pro version 10, α = 0.05, following normality testing with Shapiro-Wilk W Test. The Student’s *t*-test was used between two treatments, whereas for more than two pairs, the Tukey-Kramer honest significant difference (HSD) was applied. To explore interlinkages between variant study parameters (physical-chemical-biological), principal component analysis (PCA) was undertaken with JMP Pro 10.

## 3. Results and Discussion

### 3.1. AgNPs Characterisation: Before Testing

Characteristics of AgNPs prior to their introduction into variant exposure media are summarised in Table 1 and Appendix A. The AgNPs were near-monodispersed as their variation co-efficient was in the range of >0.05–15 [39], and there was a reasonable agreement of measured NPs size between the three techniques (TEM, NTA, and DLS). The slight discrepancies in size were attributed to differences based on principles employed to derive NPs’ size for each technique. For instance, the NTA tracks and sizes individual particles, whereas a static light path of the DLS can cause a shielding effect of smaller NPs by larger NPs and thus, size reading tends to be biassed towards larger size. AgNPs agglomerated immediately following introduction into DIW whereby 10-AgNPs exhibited higher agglomeration rates relative to 40-AgNPs; thus, indicative of higher reactivity of the former NPs due to their large surface area and higher surface atom composition [40,41]. Further, the average drift velocities for 10-and 40-AgNPs were 1687 nm/s and 1338 nm/s, respectively: hence, an additional indicator of higher reactivity of the smaller-sized NPs. In DIW, both 10- and 40-AgNPs were negatively charged at −47.93 mV, and −41.33 mV, respectively (Figure 1) associated with citrate anions influence [42], and therefore, were considered as highly stable since zeta potential was >±30 mV [43]. 

### 3.2. AgNPs Characterisation: During Exposure Period 

As depicted in Figure 1, all three water exposure regimes reduced the AgNPs’ zeta potentials relative to the DIW, and the effect was attributed to the presence of electrolytes. The neutralising effect on 10-AgNPs was more prominent in Ca^2+^, and for the 40-AgNPs in Ca^2+^ and NOM. The adsorption of media electrolytes onto AgNPs, in turn, shielded their surface charge, compressed the electric double layer (EDL), resulting in enhanced NPs agglomeration [44]. Further, NOM shielded the negative charge of citrate, thus electro-sterically improving the stability of the NPs [44,45], thereby raising their plausible enhanced persistence in the environment. Alternatively, a reduction of surface charge potential across water regimes may have also arisen from the release of Ag^+^, which may adsorb onto AgNPs surfaces, thus further exacerbating the neutralising effect on the negative charge [46]. 

The increase in agglomeration rates was linked to weakened inter-particle repulsion. For example, agglomeration of AgNPs was higher in MHW and Ca^2+^ exposure variants and relatively low in NOM due to an electrosteric stabilisation effect [15]. For the 10-AgNPs in MHW and Ca^2+^, the significant particle number increase over the exposure period was indicative of the NPs agglomeration as attested by improved particle detection using the NTA technique (Figure 2a–d, Appendix A). However, the agglomeration of 40-AgNPs (MWH and Ca^2+^) was not severe, as illustrated by relatively constant particles concentration results (Figure 3a–d). 

Notably, these findings indicate that different-sized NPs of the same parent material can experience differing degrees of physicochemical transformations dependent on water chemistry conditions, and consequently giving rise to marked differences in bio-accessibility profiles. NOM highly destabilised the 10-AgNPs (Figure 2e,f) as shown by rapid agglomeration, which plausibly induced sedimentation and particle number loss in suspension. These findings are in contrast to NOM exposures, where low agglomeration was observed due to an electrosteric stabilisation effect [15,44]. However, herein NOM influence on the stability of 40-AgNPs was not prevalent since lower agglomeration rates were exhibited, thus sustaining higher particle concentration in suspension based on results in Figure 3.

The size distributions of 10-AgNPs in NOM obtained using NTA differed markedly from other exposure regimes (Figure 2). The higher agglomeration rate of 10-AgNPs (Figure 2) compared to 40-AgNPs (Figure 3) suggested that the stabilising effect of NOM on AgNPs was size-dependent. Possibly due to their small-size driven high-reactivity, we suggest that 10-AgNPs exhibited a higher affinity for NOM, thus resulting in rapid agglomeration due to the charge shielding effect of NOM associated with their larger surface area to volume ratio [40,41]. Furthermore, the 10-AgNPs are more likely to exhibit higher collision frequency compared to 40-AgNPs. 

Initial particle concentrations were higher for 40-AgNPs at time 0 and 48 h, although the initial dosing concentration of 600 µg/L for both sizes was used (Figure 2 and Figure 3). The NTA’s lower-end detection limit ranges from 10 to 30 nm and is dependent on the refractive index of the material under question. Hence, the observed differences in particle numbers between the sizes were attributed to the equipment’s limitation to detect smaller-lower end nanoscale particles. For example, during exposure, particle concentrations for 10-AgNPs in MHW and Ca^2+^ exposures exhibited an increasing trend (due to rapid agglomeration and improved detection) and were higher compared to NOM (Figure 2). However, for the 40-AgNPs, particle concentrations in MHW and Ca^2+^ exposures showed a slight decline, increased significantly in NOM (Figure 3f) and consistent with agglomeration results derived using DLS (Appendix A). 

### 3.3. Dissolution

Herein dissolution refers to total dissolved Ag with no distinction made between different ionic species (e.g., Ag^+^, AgCl^2−^, or AgCl_3_^2−^). Dissolution was observed to be exposure water chemistry and AgNPs size-dependent (Figure 4), with the trend for 10-AgNPs in descending order being Ca^2+^ < MHW < NOM (Figure 4a). Further, a similar trend was observed for 40-AgNPs except after 24 h (Figure 4b) being MHW < Ca^2+^ < NOM. Overall, the highest dissolution rates for both AgNPs were in NOM, and least in Ca^2+^ treatments were on average 10-AgNPs (Figure 4a) were more soluble than 40-AgNPs (Figure 4b) due to the former’s larger surface area; which in turn, facilitates rapid oxidation [47]. In some instances, a slight drop in dissolution after 24 h followed by an increase after 48 h was observed. The cause for such a trend, especially a rise after 48 h, cannot be explained due to a wide 24 h analysis rate adopted herein. For future studies, more frequent dissolution analysis is recommended, for instance, every 6 h.

Furthermore, smaller sized-NPs are known to enhance particle curvature and reactivity—which in turn facilitates higher dissolution rates [48]. Notably, the relatively higher agglomeration of 10-AgNPs did not effectively lower their dissolution rate. This is probably because the formed agglomerates are not permanent but are transient where individual particles dynamically move in and out of agglomeration state [49,50], as was observed in the video recordings using NTA. Additionally, within agglomerates, microlayers of adsorbed Ag^+^, Ag^0^, and/or Ag may form and are characteristic of high dissolution rates [45]. 

Although NOM-AgNPs complexes can reduce particle surface area available for oxidation and/or inhibit dissolution; however, the coating effect does not inhibit dissolution [45]. Hence, the dissolution in NOM was plausibly enhanced by the existence of stable (preserved) AgNPs within the NOM-AgNPs complexes where a portion retained a degree of primary characteristics [51]. The stabilising effect of NOM following the formation of complexes with Au [52] and AgNPs [42] have been reported, and a similar influence of NOM on AgNPs dissolution may account for observations derived from this study. These results point to the potential NOM-facilitated persistence of NPs in aquatic environments. Concerning ionic strength, a high concentration of electrolytes (i.e., ionic strength) is known to destabilise NPs due to the adsorption of electrolytes on their surfaces [53,54]. Previously, similar transformation on size and surface characteristics of NPs have been observed and consistent with the current study findings where citrate-coated AgNPs were transformed by Ca^2+^ [39]. The high electrolyte concentration, Ca^2+^ in this case, inhibited AgNPs dissolution plausibly due to the destabilising effect linked to surface charge shielding activity.

In simple matrices such as DIW, dissolution mechanisms and thermodynamics of AgNPs are generally simple and predictable. However, in complex matrices, as in this study, AgNPs dissolution can be a highly dynamic and complex process. Nonetheless, the influence of exposure media and AgNPs size in this study was distinctively observable. Dissolution is a key determinant of AgNPs’ uptake, bioaccumulation, and effects in the aquatic environments [12,55,56]. In general, the dissolution rate of AgNPs is slow [57,58]; yet the slow and constant release of dissolved Ag may lead to high Ag bioaccumulation and toxicity effects over time [22,59]. This is because aquatic higher plants tend to have higher uptake rates for dissolved Ag compared to AgNPs [22,32]. In summary, our findings demonstrate that, as previously argued [34,60], the behaviour of NPs in actual aquatic environments cannot be fully predicted based on observations made in simple exposure media models (e.g., DIW). Hence, this necessitates detailed physicochemical characterisation of the exposure media to aid gain better insights essential to predict ENMs’ behaviour more accurately.

### 3.4. Accumulation, Uptake, and Distribution

The average whole plant Ag accumulation trend in the exposure media in ascending order was NOM < MHW < Ca^2+^ (Figure 5), with the influence of NPs size being higher in samples exposed to 10-AgNPs compared to 40-AgNPs (Appendix A). Further, overall Ag accumulation was higher in roots relative to fronds (Figure 6), as similarly observed by Souza et al. [17]. The influence of exposure media chemistry on the interaction of NPs with plants remain amongst the understudied aspects that may influence uptake processes by biota [20]. For example, Glenn and Klaine [52] reported that NOM in exposure media reduced Au accumulation in plants exposed to AuNPs. This is due to the likely formation of large-sized NOM-NPs complexes that may hinder the adsorption and internalisation potential of dissolved and particulate forms of Ag. Hence, such a mechanism may limit Ag accumulation, and we postulate to similarly account for the observations derived from this study.

However, no link between the accumulated Ag and hydrodynamic sizes of AgNPs across the exposure regimes was observed. For instance, Ag accumulation was generally higher in 10-AgNPs irrespective of water chemistry type, although they generally had formed larger agglomerates. Therefore, our findings are consistent with reports that higher plants tend to rapidly internalise ionic species relative to AgNPs [17,24,33]—and a behaviour seemingly applicable to metal-based ENMs [12]. However, results of Ag dissolution vs accumulation as influenced by media chemistry showed an inverse trend with dissolution being least in Ca^2+^ variant, yet accumulation was higher under similar exposure conditions. The basis for such a relationship could not be established. Thus, the underpinning mechanism on how NPs interact with plants as influenced by water chemistry remains an open scientific enquiry.

The observed trend of AgNPs’ size influence on plants Ag accumulation bears similarity to earlier works on AuNPs [23,52], where Au plant accumulation was relatively higher from exposure with smaller NPs compared to larger-sized ones. In the current study, we propose that the observed trend was due to the relatively higher dissolution of 10-AgNPs as well as their enhanced adsorption to plant surfaces. The latter was informed by the averagely higher collision frequency of 10-AgNPs relative to 40-AgNPs as observed using NTA.

Accumulation of Ag was observed to be higher in roots compared to fronds (Figure 6), plausibly an indication of roots being the major sites for AgNPs/dissolved Ag uptake. Again, this finding is consistent with earlier works where relatively higher deposition of metals was observed in the roots of aquatic higher plants compared to fronds following exposure to metal-based NPs [23,30,52,53]. In contrast, Hu et al. [27] observed a higher accumulation of Zn in fronds of *S. minima* exposed to ZnONPs; but the underlying cause or mechanism for enhanced accumulation of Zn in the fronds was not explained. In comparison to AgNPs used herein, we postulate that this was probably due to differences in physiological uptake pathways of Ag and Zn as the latter is an essential nutrient to plants, unlike the former. Although *Salvinia* roots are modified frond elongation [61], they function as normal roots because the presence of numerous root hairs dramatically increases surface area for absorption and adsorption of metals. In turn, this points to why roots are more efficient in nutrient uptake compared to fronds. This is because roots are fully immersed in water, unlike fronds; thus, having enhanced potential for adsorption or uptake of AgNPs.

To ascertain whether AgNPs were internalised by *S. minima*, studies on EDS coupled SEM analysis on roots revealed poor Ag signals (<1% weight) on the root surfaces for adsorbed Ag (Appendix A). This implied limited likelihood for the successful detection of internalised AgNPs using TEM; hence, no further attempts were made to investigate internalisation with TEM. Although internalised NPs have successfully been visualised using electron microscopy techniques in aquatic higher plants [32,62]; however, this was only at several-fold higher exposure concentrations (for instance, up to 50 mg/L [62] and 50 mg/L [32]) compared to this study.

Biota tissue examination with electron microscopy is highly complex and consequently does not yield definitive and conclusive evidence for NPs internalisation. For instance, sample representativeness for biota as large as the specimen used in this study raises the likelihood of missing internalised NPs for two reasons. First, because it is impractical to analyse all tissue locations, and secondly, due to the non-homogeneous distribution of NPs in the specimen. Other challenges in determining the internalisation of NPs in biota tissues using imaging-based techniques are the false-negative findings either as a result of artefacts linked to stains and buffers used during sample preparation or the presence of electron-dense biological nanoscale matter [20]. Overall, herein we postulate that owing to size ranges of AgNPs used in this study, there was a likelihood for internalisation since larger sized NPs agglomerates were observed to be internalised [32], with smaller sizes at higher rates compared to larger counterparts [23].

The principal component analysis (PCA) results suggested a high probability for the internalisation of both particulates and dissolved Ag forms (Appendix A) but were inconclusive on the dominant principle driving parameter between AgNPs size and dissolution. Despite this limitation, PCA results suggest plausible multi-pathways on the accumulation of Ag influenced by the interplay between AgNPs size and water chemistry.

Efforts to visualise the distribution of AgNPs/Ag*X* in exposed plants using the XRF image analysis technique were unsuccessful, even after the dosing concentration was increased to 2 mg/L (results not shown). Previous XRF successes in examining the distribution of AgNPs/Ag*X* in higher plants were achieved at excessively higher dosing concentrations of up to 40 mg/L [26,29,63], and as such, are deemed environmentally irrelevant and probably were utilised primarily to achieve analytical detection. At such high dosing concentrations, plants can experience disruption of epidermal protective barriers and loss of osmoregulatory control that is unlikely to occur at low concentrations [25,32,64]. Our findings highlight weak capability by XRF to detect Ag distribution in plants exposed to concentration ranges used herein (≤600 µg/L), and remarkably from those anticipated in actual environments as are very low in the order of ng/L to µg/L [10,65,66,67]. In this context, our study findings bring into sharp focus the need to develop appropriate analytical tools with capabilities to characterise NPs in complex environmental and biota matrixes.

### 3.5. Plant Growth and Chlorophyll Pigments

Results in Figure 7 illustrate AgNPs induced growth inhibition effects to *S. minima* dependent on NPs size and exposure media chemistry. In MHW, 10-AgNPs induced higher growth inhibition relative to 40-AgNPs. However, an opposite effect in NOM was observed where 40-AgNPs exhibited higher growth inhibition. In Ca^2+^, the inhibition effect for both NPs sizes was relatively similar, although smaller-sized counterparts were slightly more toxic and severe relative to other exposure variants. Further, 10-AgNPs exerted higher growth inhibition except under NOM treatment (Appendix A), possibly due to harmful effects of Ca^2+^ at excessive concentrations above essential levels [68].

Exposure to AgNPs also induced alterations to chlorophyll *a* and *b* content, and the effects were non-uniform across different water chemistries and size-dependent (Appendix A). In MHW, both sizes significantly reduced the chlorophylls relative to controls—with 10-AgNPs being most inhibitive. In NOM, 10-AgNPs insignificantly reduced the chlorophylls; but no effects were observed following exposure to 40-AgNPs. In contrast, under Ca^2+^ exposure 40-AgNPs exerted highly inhibitive effects (Appendix A), whereas 10-AgNPs stimulated chlorophyll production levels higher relative to the controls. Jiang et al. [16] reported that AgNPs exposure concentrations of ≥5 mg/L inhibited Chl *a* and *b* in *Spirodela polyrhiza.* Moreover, inhibition of chlorophyll pigments and carotenoids had the likelihood of reducing the plants’ ability to cope with increasing UV irradiation stress [16]. Therefore, adverse effects on photochemical efficiency (energy conversion and housing) of plants point to possible secondary effects on numerous active (energy-requiring) physiological pathways, for instance, such as detoxification of environmental xenobiotics and plant growth.

Results herein demonstrate that the higher rate of agglomeration for 10-AgNPs had no influence on their toxicity potential as they were more toxic compared to 40-AgNPs (Figure 7). This aspect was attributed to dissolution facilitated effects. For *L. minor* growth: larger AgNPs (84.1 nm) were more growth inhibitive (EC_50_ = 61.64 µg/L) after 7 d compared to that of smaller forms (15.92 nm: EC50 = 63.71 µg/L); but an opposite trend was observed after 14 d [69]. It is probable that the initial high agglomeration of smaller-sized AgNPs may have hindered their dissolution-mediated toxicity. However, over elongated exposures may result in physicochemical transformations (e.g., size, surface properties, and dissolution), which alter the bioavailability and toxicity of the NPs, for instance, as shown in longer 14 d experiments by others [12,69].

## 4. Conclusions

The current findings demonstrate the significant interactive influence of inherent ENMs’ physicochemical characteristics and the exposure water chemistry on the behaviour and exposure potential of ENMs in aquatic systems. The AgNPs’ aqueous behaviour, bioaccumulation and toxicity profiles were distinct between the two sizes and exposure water chemistry; illustrating that exposure assessments of nanopollutants in water resources need not only be based on predictions/observations of ENMs pristine characteristics but require adequate consideration of water physicochemical characteristics of the site being examined in pursuit of drawing meaningful conclusions from the derived results. The 10-AgNPs were considerably more soluble and toxic than 40-AgNPs, and higher Ag accumulation was from samples exposed to the former.

Dissolution was a key driver for Ag accumulation and toxicity of the AgNPs to *S. minima*. The results demonstrated the size-driven differences in behaviour and toxicity, thus strengthening the case for the need to develop and adopt unique approaches for ENMs’ environmental risk assessment as opposed to current paradigms applied for bulk counterpart chemicals.

In cases with low concentrations similar to those examined herein, ICP-MS was found to be a valuable tool to quantify adsorbed and/or absorbed ionic or particulates forms of the metal-based ENMs following their interactions with higher aquatic plants. Because ICP-MS determines the total content, hence it cannot provide information on the mode of accumulation as either adsorption or absorption (internalisation). Broadly, imaging techniques provide meaningful results (in terms of adsorption or absorption) at significantly high and environmentally irrelevant exposure concentrations; however, they are time-consuming and prone to interferences by biota components—thus raising the probability for misinterpretation of results (false negatives).

Considering the persistent question of ENMs’ internalisation potential by aquatic higher plants and current limited analytical capability, we propose further efforts be expended towards characterising the physicochemical properties of ENMs in situ as means of basing predictions on their bio-accessibility potential, or lack thereof. For instance, information on size distributions as well as the ratio of particulates vs dissolved components of ENMs in a given exposure media can provide valuable insights on possible exposure dynamics and likelihood for ENMs internalisation. We, therefore, highlight the need to re-examine the increasing quest for internalisation information when bioaccumulation results confirm the association of ENMs with plants, especially in the context of water resources pollution and ecological health in general. Based on the current analytical challenges, models can still be drawn from high exposure concentrations, which enable visualisation of ENMs interaction biota.

*S. minima* and similar aquatic higher plants exhibit characteristics of being reserves of ENMs in aquatic systems and can transfer nanopollutants to higher trophic levels. The observed ability of such plants to bioaccumulate ENMs or by-products strengthen support for calls to manipulate aquatic higher plants for phytoremediation purposes.

## Figures and Tables

**Figure 1 molecules-26-02305-f001:**
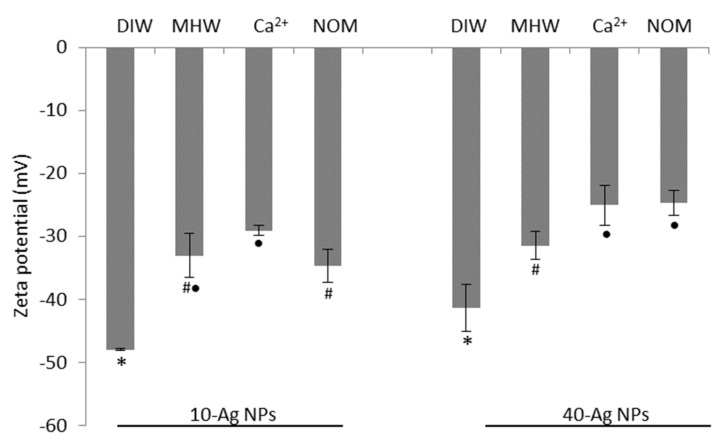
Zeta potentials of silver nanoparticles (AgNPs) in DIW at 0 h, and in water exposure regimes of moderately hard water (MHW), water containing natural organic matter (NOM), and Ca^2+^ after 48 h. Bars denote standard error (*n* = 3) andthe symbols below the bars (*, #, •) represent statistical difference where similar symbols are indicative of no statistical diference whereas differing symbols indicate statistical difference between water treatments within a specific AgNPs size as tested at *p* < 0.05 with Turkey Kramer honest significant difference (HSD).

**Figure 2 molecules-26-02305-f002:**
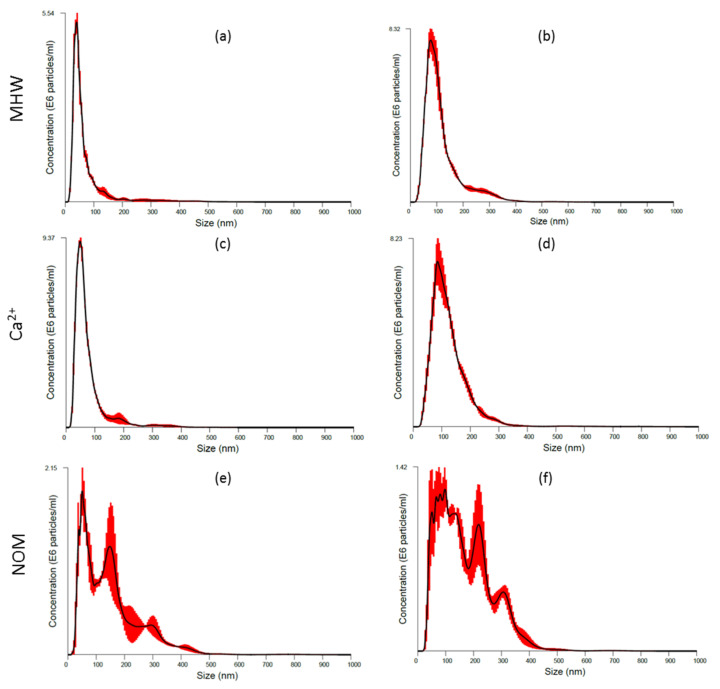
Size distributions obtained from NTA for 10-AgNPs in MHW: (**a**) 0 h, (**b**) 48 h; in Ca^2+^: (**c**) 0 h, (**d**) 48 h; and in NOM: (**e**) 0 h, (**f**) 48 h. Red bars indicat e standard error (*n* = 3).

**Figure 3 molecules-26-02305-f003:**
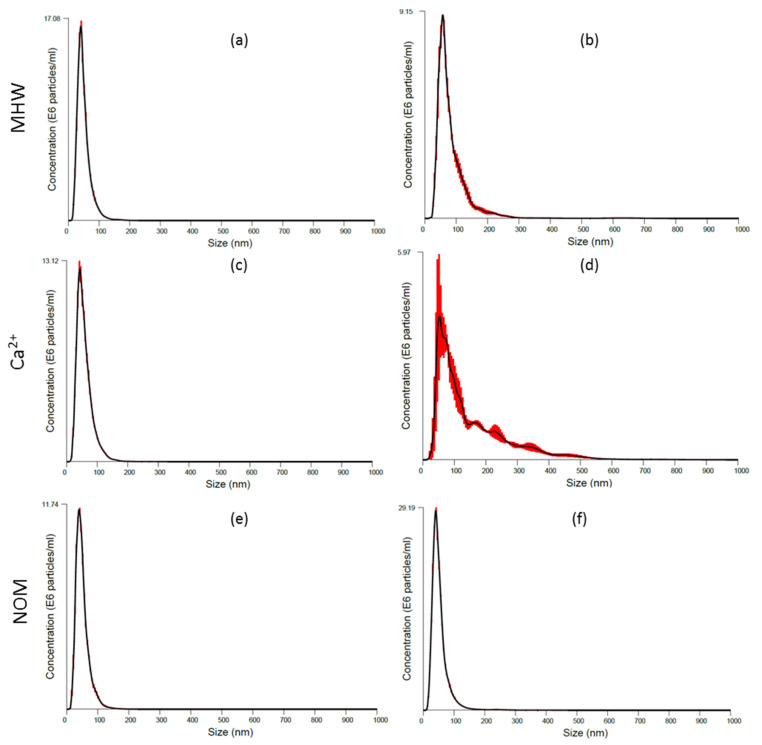
Size distribution obtained from NTA for 40-AgNPs in MHW: (**a**) 0 h, (**b**) 48 h; in Ca^2+^: (**c**) 0 h, (**d**) 48 h; and in NOM: (**e**) 0 h, (**f**) 48 h. Red bars indicate standard error (*n* = 3).

**Figure 4 molecules-26-02305-f004:**
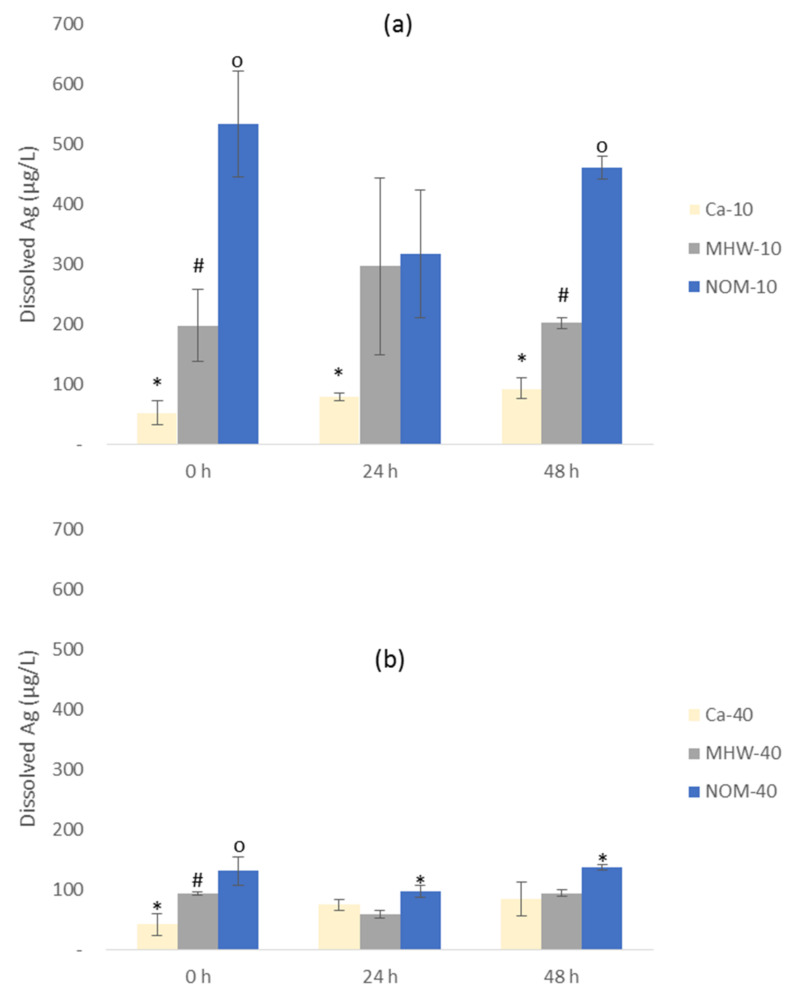
Dissolution of AgNPs at 0, 24 and 48 h in MHW, NOM and Ca^2+^ water exposures for: (**a**) 10-AgNPs and (**b**) 40-AgNPs. Bars denote standard error (*n* = 3) and the symbols on top (*, #, o) represent statistical difference where similar symbols are indicative of no statistical diference whereas differing symbols indicate statistical difference between water treatments within a specific AgNPs size as tested at *p* < 0.05 with Turkey Kramer honest significant difference (HSD).

**Figure 5 molecules-26-02305-f005:**
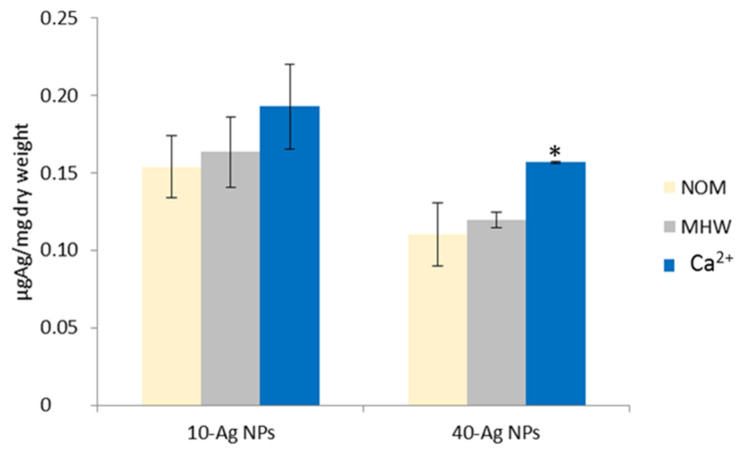
Pairwise comparison of whole plant Ag accumulation across water quality treatments within a single AgNPs size exposure (MHW, NOM, and Ca^2+^). The * symbol depicts statistical diference (*n* = 3) of Ca^2+^ from other media samples within 40-AgNPs size, tested at *p* < 0.05 with Turkey Kramer HSD.

**Figure 6 molecules-26-02305-f006:**
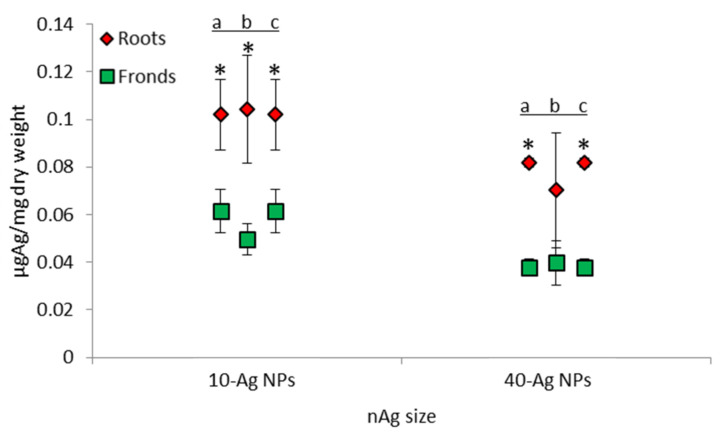
Accumulation of Ag by *Salvinia minima* roots and fronds after 48 h for the 10- and 40-AgNPs obtained from inductively coupled plasma mass spectroscopy (ICP-MS) analysis. Bars denote standard error (*n* = 3) and the * indicates a statistical difference between roots and fronds Ag accumulation within a specific water type; a = MHW, b = NOM, and c = Ca^2+^. Student’s *t*-test, *p* < 0.05: MHW (10-AgNPs = 0.022, 40-AgNPs = 0.0003), NOM (10-AgNPs = 0.043, 40-AgNPs = 0.147), Ca^2+^ (10-AgNPs = 0.039, 40-AgNPs = 0.0001).

**Figure 7 molecules-26-02305-f007:**
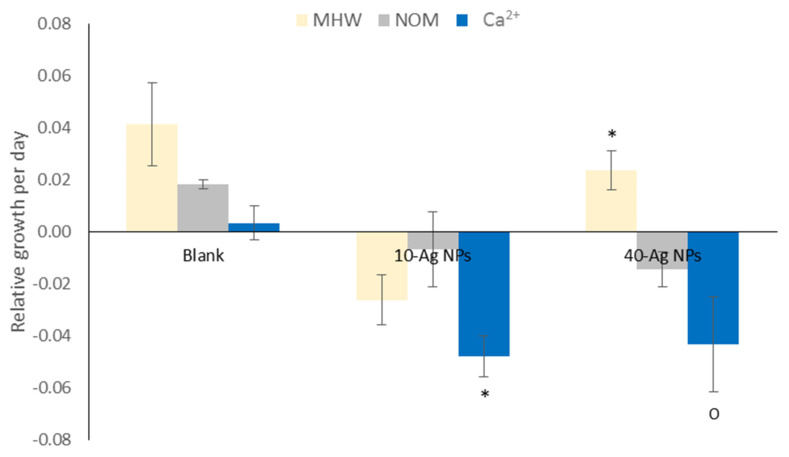
Relative growth rates of *S. minima* with 10-AgNPs and 40-AgNPs in MHW, NOM and Ca^2+^ exposures after 48 h. Bars denote standard error (*n* = 3) and the signs (*, o) associated with bars indicate differing growth rate between AgNPs exposures within water treatment. Turkey Kramer HSD, *p* < 0.05.

**Table 1 molecules-26-02305-t001:** The characteristics of silver nanoparticles (AgNPs), 10-AgNPs and 40-AgNPs, in deionised water (DIW) obtained using different techniques. Data are presented as averages (mode for nanoparticle tracking analysis (NTA)), and in the parentheses are the data spread and standard deviations.

	TEM (nm)	NTA (nm)	Zetasizer (nm)	ζ Potential (mV)
10-AgNPs	8.6 (±2.12; 0.24)	34.6 (±3.4; 0.09)	40 (±5.6; 0.14)	−47.93
40-AgNPs	41.45 (±4.57; 0.11)	47.67 (±1.33; 0.02)	56.71 (±3.9; 0.06)	−41.33 mV

## Data Availability

The data presented in this study are available on request from the corresponding author.

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
