# Peer review of "Exposure Media and Nanoparticle Size Influence on the Fate, Bioaccumulation, and Toxicity of Silver Nanoparticles to Higher Plant Salvinia minima"

_molecules, 2021, doi:10.3390/molecules26082305_

Round 1
Reviewer 1 Report
Basically, the approach taken in the study is interesting, and data may be appropriate for publication in Molecules. I have some questions and comments:
1. Why was the dose of AgNPs 600 µg/L used in the experiments? On what basis was this value chosen? (section 2.2.)
2. What does the abbreviation ENPs stand for in section 2.3?
3. Please provide the parameters of irradiating plant cultures.
4. It is not entirely clear how long the experiments were conducted. In section 2.5 it says 2 days, but in section 2.8 it says 7 days. If plant exposures to AgNPs were tested for 2 days, why was the exposure time selected this way?
5. In Figure 1 the abbreviations (DIW, MHW etc) should be explained.
6. Conclusions are too long. Conclusions should be short and concise.
Author Response
Review Comment 1.1: Why was the dose of AgNPs 600 µg/L used in the experiments? On what basis was this value chosen? (section 2.2.)
Response 1.1: Our previous study (Thwala et al. 2013) and others in general have utilized excessive exposure dosages ranging from 10-1000 mg/L, which are not environmentally realistic. In order to improve environmental realism whilst attaining analytical success; the 600 µg/L dosage was selected. At some of the highest predicted environmental concentrations of ~ 2.63 µg/L, it would have been analytically challenging to examine biota interactions.
Review Comment 1.2: What does the abbreviation ENPs stand for in section 2.3?
Response 1.2: changed to AgNPs for clarity
Review Comment 1.3: Please provide the parameters of irradiating plant cultures.
Response 1.3: Provided in Section 2.4, exposures were undertaken under similar conditions.
Review Comment 1.4: It is not entirely clear how long the experiments were conducted. In section 2.5 it says 2 days, but in section 2.8 it says 7 days. If plant exposures to AgNPs were tested for 2 days, why was the exposure time selected this way?
Response 1.4: The 2.8 exposures were extended to 7 days as no growth changes could be observed within 48 h during initial optimising experimental design runs. Others were fixed at 48 and 72 hrs as dynamics could be detected during that test duration.
Review Comment 1.5: In Figure 1 the abbreviations (DIW, MHW etc) should be explained.
Response 1.5: The abbreviations were explained on first mention in sections 2.1 and 2.2.
Review Comment 1.6: Conclusions are too long. Conclusions should be short and concise.
Response 1.6: The authors reviewed the Conclusions but hold the view that significant text reduction may limit the message being communicated. Furthermore, being five paragraphs of a few lines the section is not considered unnecessarily elongated.
Reviewer 2 Report
The authors provide detailed information on preparation, storage and characterisation.
Several comments were made in text, please view attached file. Also an incomplete sentence was spotted, please correct this.
Overall this is interesting work and raises questions to the protocols that should be put in place for measurements and protocols.
p.s. please ignore any highlighting, it is meant for personal use.

Author Response
All the grammatical/editorial review comments were accepted.
Review Comment 2.1: Why this pH value?
Response 2.1: This was done to standardize the pH across all exposure solutions (MHW, Ca, NOM) as initially were either just below or above the pH of 8. Explanation has been provided in text as well.
Review Comment 2.2.1: For the 10 nm NPs, 3 different instruments provide 8.6, 34.6, and 40 nm. The needs discussion.
Response 2.2.1, explanation in text expanded: The slight discrepancies in size were attributed to differences based on principles employed to derive NPs’ size for each technique. For instance, the NTA tracks and sizes individual particles, whereas a static light path of the DLS can experience a shielding effect of smaller NPs by larger NPs and thus size reading tends to be biased towards larger size.
Review Comment 2.2.2: The zetasizer value for 40 nm NPs also needs commenting.
Response 2.2.2: The 10 and 40 nm surface charge potentials originally differed as per manufacturer specifications. So the authors are unable provide valuable further comments on this aspect.
Review Comment 2.3: Please comment on the fact that 0 and 48 h seems to be on the same levels, while 24 h drops.
Response 2.3, in text provided: In some instances, a slight drop in dissolution after 24 h, followed by an increase after 48 h was observed. The cause for such a trend, especially the rise after 48 h, cannot be explained due to a wide 24 h analysis rate adopted. For future studies, more frequent monitoring is recommended, for instance, every 6 h.
Review Comment 2.4: Use of the PCA method here is risky as by nature PCA components are indirectly translated. It is not clear how authors arrive to the conclusions mentioned.
Response 2.4: The purpose of running a PCA was to examine relativeness, specifically dissolution and Ag accumulation in this instance. The relativeness between the two parameters was used as basis for possible internalisation of Ag ions amongst contributors to Ag accumulation. In the absence of measurements the authors only raised it as highly possibility but did not suggest definite occurrence, but others [29-31] point to high likelihood for internalisation of dissolved ions.
Review Comment 2.5: incomplete sentence
Response 2.5_sentence completed in text: The 10-AgNPs were considerably more soluble and toxic than 40-AgNPs, and higher Ag accumulation was from samples exposed to the former.